# Comparison of the Drug-Induced Efficacies between Omidenepag Isopropyl, an EP2 Agonist and PGF2α toward TGF-β2-Modulated Human Trabecular Meshwork (HTM) Cells

**DOI:** 10.3390/jcm11061652

**Published:** 2022-03-16

**Authors:** Soma Suzuki, Masato Furuhashi, Yuri Tsugeno, Araya Umetsu, Yosuke Ida, Fumihito Hikage, Hiroshi Ohguro, Megumi Watanabe

**Affiliations:** 1Departments of Ophthalmology, School of Medicine, Sapporo Medical University, Chuo-ku, Sapporo 060-8556, Japan; ophthalsoma@sapmed.ac.jp (S.S.); yuri.tsugeno@gmail.com (Y.T.); araya.umetsu@sapmed.ac.jp (A.U.); funky.sonic@gmail.com (Y.I.); fuhika@gmail.com (F.H.); ooguro@sapmed.ac.jp (H.O.); 2Departments of Cardiovascular, Renal and Metabolic Medicine, Sapporo Medical University, Sapporo 060-8556, Japan; furuhasi@sapmed.ac.jp

**Keywords:** 3D spheroid cultures, human trabecular meshwork, omidenepag isopropyl, omidenepag, PGF2α, TGFβ2

## Abstract

To compare the drug-induced efficacies between omidenepag (OMD), an EP2 agonist, and prostaglandin F2α (PGF2α) on glaucomatous trabecular meshwork (TM) cells, two- and three-dimensional (2D and 3D) cultures of TGF-β2-modulated human trabecular meshwork (HTM) cells were used. The following analyses were performed: (1) transendothelial electrical resistance (TEER) and FITC-dextran permeability measurements (2D), (2) the size and stiffness of the 3D spheroids, and (3) the expression (both 2D and 3D) by several extracellular matrix (ECM) molecules including collagen (COL) 1, 4 and 6, and fibronectin (FN), and α smooth muscle actin (αSMA), tight junction (TJ)-related molecules, claudin11 (Cldn11) and ZO1, the tissue inhibitor of metalloproteinase (TIMP) 1–4, matrix metalloproteinase (MMP) 2, 9 and 14, connective tissue growth factor (CTGF), and several endoplasmic reticulum (ER) stress-related factors. TGF-β2 significantly increased the TEER values and decreased FITC-dextran permeability, respectively, in the 2D HTM monolayers, and induced the formation of downsized and stiffer 3D HTM spheroids. TGF-β2-induced changes in TEER levels and FITC-dextran permeability were remarkably inhibited by PGF2α. PGF2α induced increases in the sizes and stiffness of the TGF-β2-treated 3D spheroids, but OMD enhanced only spheroid size. Upon exposure to TGF-β2, the expression of most of the molecules that were evaluated were significantly up-regulated, except some of ER stress-related factors were down-regulated. TJ-related molecules or ER stress-related factors were significantly up-regulated (2D) or down-regulated (3D), and down-regulated (2D) by PGF2α and OMD, while both drugs altered the expression of some of the other genes in the 3D spheroids in a different manner. The findings presented herein suggest that PGF2α and OMD differently modulate the permeability of the TGFβ2-modulated 2D monolayers and the physical properties of the 3D HTM spheroids.

## 1. Introduction

In terms of the etiology of glaucoma [1,2], it is well known that elevated levels of the intraocular pressure (IOP) are critical factors [3], and therefore, hypotensive therapies by the use of anti-glaucoma medications in conjugation with laser or surgical intervention is recognized as the only currently available evidence-based therapy [4]. Physiologically, IOP levels of the human eye are maintained by aqueous humor (AH) dynamics, that is, AH production and drainage occur mainly through the trabecular meshwork (TM) pathway [5]. The cause of such an IOP elevation is generally thought to be due to an elevated resistance to AH drainage through the TM pathway presumably by the excess depositions of several extracellular matrix (ECM) molecules [6]. It has been reported that such an excess of ECM deposits of TM may be mediated by elevated AH levels of transforming growth factor (TGF)-β2 [7]. In fact, an in vitro study using a rat model indicated that TGF-β2 caused an elevation in IOP levels by increasing the resistance to AH outflow [8]. As a result, TGF-β2 is now commonly used as an in vitro model using the TM cell cultures [9].

In addition to the first-line drugs, prostaglandin (PG) F2α agonists (PGF2α-ags) are used in the treatment of patients with glaucoma and ocular hypertension (OH) [10,11,12], omidenepag isopropyl (OMDI), a non-prostaglandin, a prostanoid EP2 agonist, has recently become available [13,14]. Pharmacologically, (1) OMDI is metabolized into the active form (omidenepag, OMD) by hydrolysis after its administration, and (2) the hypotensive mechanisms for OMD involve an increase in the ease of uveoscleral outflow, similar to PGF2α-ags [15]. However, a recent in vivo study reported that the pharmacokinetics are quite different for OMD compared to PGF2α-ags [16]. In addition, the effects of these drugs on the ease of conventional outflow have not been extensively characterized at this time, since an appropriate experimental model for evaluating the drug efficacies on conventional versus uveoscleral roots has not been available. To address this difficulty, we recently succeeded in preparing a 3D human TM (HTM) spheroid as a suitable in vitro model [17], in which no assistance, such as a collagen-gel or a scaffold that is usually used in the 3D culture method [18], was required.

Therefore, in the current study, to compare the roles between PGF2α-ags and an EP2 agonist in glaucomatous HTM, we used TGF-β2-treated 2D and 3D HTM cells and subjected them to the following analyses; (1) the transendothelial electrical resistance (TEER) and FITC-dextran permeability (2D), (2) the physical properties of 3D spheroids, including size and stiffness, and (3) the expression of major extracellular matrix (ECM) molecules including collagen (COL) 1, 4 and 6, fibronectin (FN) and α smooth muscle actin (αSMA), tight junction (TJ) related molecules, claudin11 (Cldn11) and ZO1, tissue inhibitors matrix proteinase (TIMP) 1–4, matrix metalloproteinase (MMP) 2, 9 and 14, and connective tissue growth factor (CTGF), and endoplasmic reticulum (ER) stress-related factors (2D and 3D).

## 2. Materials and Methods

### 2.1. Human Trabecular Meshwork (HTM) Cells

All experiments using human-derived cells were conducted in compliance with the tenets of the Declaration of Helsinki after approval by the internal review board of Sapporo Medical University. Immortalized HTM cells transfected with an original defective mutant of the SV40 virus were purchased from Applied Biological Materials Inc., Richmond, Canada, and these cells were characterized as described in the consensus recommendations for TM cells [19] in advance of the current study described below.

### 2.2. 2D and 3D Spheroid Cultures of Human Trabecular Meshwork (HTM) CELLS

The 2D-cultured HTM cells were further processed to obtain 3D spheroids in a hanging droplet culture plate (# HDP1385, Sigma-Aldrich Co., St. Louis, MO, USA) during a period of 6 days as described in our previous report [17] in the absence or presence of 5 ng/mL TGF-β2 and/or either 100 nM PGF2α or Omidenepag (OMD). The dosages of these TGF-β2 and PG derivatives were identical to those used in our previous studies [17,20,21].

### 2.3. Transendothelial Electron Resistance (TEER) and FITC Dextran Permeability Measurements of 2D HTM Cultured Monolayer

Barrier functions of the 2D HTM cell monolayers were evaluated by TEER and FITC-dextran permeability measurements, as described previously using 12-well plates for TEER (0.4 μm pore size and 12 mm diameter; Corning Transwell, Sigma-Aldrich) and the TEER values (Ωcm^2^) were recorded by an electrical resistance system (KANTO CHEMICAL Co., Inc., Tokyo, Japan) [17,22]. Concerning FITC-dextran permeability, the concentrations of the FITC-dextran that penetrated through the 2D monolayer during a period of 60 min were determined using a multimode plate reader (Enspire; Perkin Elmer, MA, USA).

### 2.4. Physical Properties Measurements, including the Size and Stiffness, of 3D Spheroids

The physical properties, size, and stiffness of the 3D HTM spheroids were evaluated by measurements of the largest cross-sectional area (CSA) from phase-contrast images and the force (μN/μm) required to induce a 50% deformation during a 20 s period by a micro-squeezer (MicroSquisher, CellScale, Waterloo, ON, Canada), respectively, as previously reported [17,23].

### 2.5. Quantitative PCR

Quantitative PCR was performed using the specific primers (Appendix A), and fold-changes in the normalized housekeeping gene 36B4 (Rplp0) were calculated as described previously [17]. Primers’ information and running protocols were described in Appendix A, respectively.

### 2.6. Immunofluorescent Labeling

Immunofluorescent labeling of 2D- and 3D-cultured HTM was performed by a previously described method. Briefly, after blocking in 3% BSA in PBS, the 4% paraformaldehyde (PFA) fixed 2D-cultured cells or 3D spheroids were successively incubated with (1) a primary antibody (1:200 dilution) including anti-collagen 1, 4 or 6 (#600-401-103-0.1, #600-401-106-0.1, #600-401-108-0.1, ROCKLAND antibodies & assays, Limerick, PA, USA), anti-fibronectin (#sc-8422, Santa Cruz Biotechnology, INC. Dallas, TX, USA) or anti-αSMA (#ab5694, Abcam, Cambridge UK) details of these primary antibodies are listed in Appendix A), and (2) an Alexa Flour 488 labeled secondary antibody (1:500 dilution), and phalloidin (#A12379, ThermoFisher, Waltham, MA, USA) and DAPI (1:1000 dilution) (#D523, Doujin, Japan). Their immunofluorescent images were obtained using a Nikon A1 confocal microscope and NIS-Elements 4.0 software as described previously [17].

### 2.7. Statistical Analysis

All statistical analyses were performed using Graph Pad Prism 9 (GraphPad Software, San Diego, CA, USA), and statistical significance was determined by a confidence level greater than 95% in a two-tailed Student’s *t*-test or two-way analysis of variance (ANOVA) followed by a Tukey’s multiple comparison test was performed as described previously [17].

## 3. Results

To compare the effect of PGF2α and an EP2 agonist, omidenepag (OMD), on glaucomatous human TM, 2D- and 3D-cultured HTM cells that had been treated with TGF-β2 (5 ng/mL) [17] were used. Initially, to study the barrier function of the 2D HTM cell monolayers, transendothelial electron resistance (TEER) and FITC-dextran permeability measurements were performed. As shown in Figure 1, upon treatment with a 5 ng/mL solution of TGF-β2, the TEER values were substantially increased and a decrease in FITC-dextran permeability was observed, and these TGF-β2 induced changes were significantly suppressed by PGF2α, but not by OMD. This finding suggests that such suppressive effects toward the barrier function in the TGF-β2-treated 2D HTM monolayers are exclusively FP2 dependent.

We next examined the drug-induced effects of PGF2α and OMD on TGF-β2-treated 3D HTM spheroids with respect to their physical properties, size, and stiffness. As shown in Figure 2, the sizes of the TGF-β2 untreated control 3D HTM spheroids became substantially decreased during the 6-day culture, and these downsizing effects were further enhanced in the case of the TGF-β2-treated spheroids at days 3 and 6. Furthermore, the TGF-β2-treated 3D HTM spheroids became significantly larger at day 6 in the presence of either PGF2α or OMD. Alternatively, the administration of TGF-β2 significantly induced an increased stiffness in the 3D HTM spheroids, and the TGF-β2-induced effect was further increased by the presence of PGF2α, but not OMD (Figure 3).

To elucidate the underlying mechanisms responsible for the above effects by PGF2α and OMD on the TGF-β2-treated 2D- and 3D-cultured HTM cells, the expression of ECM proteins (*COL1, 4*, and *6, FN*, and α*SMA*) was estimated by qPCR analysis and immunolabeling. The administration of TGF-β2 induced a significant up-regulation in the mRNA expressions of all ECMs molecules (2D), and *COL1, COL6*, and *FN* (3D). The addition of PGF2α or OMD induced a significant up-regulation in *COL4* (2D) and a down-regulation in *COL1* (3D), *COL6* (3D), *FN* (3D), and α*SMA* (3D), and a substantial up-regulation of *COL4* (2D) and down-regulation of *COL6* (3D) and *FN* (3D), respectively (Figure 4). Immunolabeling of 2D-cultured HTM cells with specific antibodies was nearly identical to those for gene expression (Appendix A). However, in contrast, the immunolabeling results of the 3D spheroids indicated that the expression of COL6 was relatively up-regulated and significantly down-regulated by PGF2α or OMD similarly to mRNA expression (Figure 5). The expressions of the other ECM molecules were not significantly modulated among the 3D HTM spheroid experimental groups (Appendix A). In terms of these discrepancies between qPCR and immunolabeling of the 3D spheroids, this was also recognized in our previous study using HTM cells [17,24] as well as other sources of cells [25,26,27]. As a possible reason for this, we speculate that immunolabeling may adequately reflect the expression of the target molecules that are located on the surface of the 3D spheroids, in contrast to their total expressions detected by qPCR analysis.

To examine this issue further, qPCR analyses of tight junction (TJ)-related proteins in TM, claudin11 (Cldn11) and ZO1 [28,29], ECM regulatory factors; *TIMP1-4*, *MMP2, 9* and *14*, and *CTGF*, and ER stress-related factors; the glucose regulator protein (GRP)78, GRP94, the X-box binding protein-1 (XBP1), spliced XBP1 (sXBP1) and the CCAAT/enhancer-binding protein homologous protein (CHOP) were performed. TJ-related molecules and the mRNA expression of Cldn11 and ZO1 were significantly up-regulated (2D) or down-regulated (3D) by the presence of PGF2a or OMD although the values were not affected by TGF-β2 (Figure 6). While in contrast, *TIMP2* (2D), *TIMP3* and *4* (3D), *MMP2*, *9* and *MMP14* (2D and 3D), *CTGF* (2D and 3D), and all five ER stress-related genes except *GRP78* (2D) or *GRP94*, *XBP*, and *sXBP* (3D) were significantly up-regulated or down-regulated, respectively, upon the administration of a 5 ng/mL solution of TGF-β2 (Appendix A). The addition of PGF2α or OMD induced a significant down-regulation of *TIMP1* (2D and 3D), *TIMP3* (3D), *MMP14* (3D), GRP 78 (2D), GRP 94 (2D), XBP (2D), sXBP (2D), and CHOP (2D and 3D) and an up-regulation of CTGF (2D) and a down-regulation of *TIMP3* (3D) as well as all five ER stress-related genes (2D), and an up-regulation of *CTGF* (2D), respectively (Appendix A). The above observations are summarized in Table 1; (1) TGF-β2 induced a significant up-regulation of most of the ECM proteins, ECM regulatory factors, TIMPs and MMPs, and CTGF in both 2D- and 3D-cultured HTM cells, but the expression of ER stress-related was different between the 2D- (up-regulated) and the 3D (down-regulated)-cultured HTM cells, and (2) PGF2α and OMD had similar effects on the expressions of *COL4* (2D), *COL6* and *FN* (3D), TJ-related molecules (2D; up-regulation, 3D; down-regulation), *TIMP3* (3D; up-regulation), *CTGF* (2D; up-regulation) and most of the ER stress-related factors, but some gene expressions including for *COL1* and α*SMA* (3D), *TIMP1* (2D and 3D)*, MMP9* (3D), and *CHOP* (3D) were different between two drugs. Therefore, these collective findings suggest that the PGF2α and OMD induced the up-regulation of TJ-related molecules (2D) and their diverse effects on the expression of several genes may cause diverse effects toward the barrier functions of the 2D HTM monolayers as well as the physical properties of the 3D HTM spheroids, as described above.

At day 6, HTM 2D cells and 3D spheroids (NT: non-treated control) and those treated with a 5 ng/mL solution of TGF-β2 (TGFβ) in the absence and presence of 100 nM PGF2α (PG) or the EP2 agonist, omidenepag (OMD), were subjected to qPCR analysis to estimate the expression of mRNA in ECMs (*COL1*, *COL4*, *COL6*, *FN*, and α*SMA*). Analyses were performed in triplicate using fresh preparations (*n* = 12–15 3D spheroids each). Data presented are the arithmetic mean ± standard error of the mean (SEM), and statistical differences were determined by ANOVA followed by a Tukey’s multiple comparison test. * *p* < 0.05, ** *p* < 0.01, *** *p* < 0.005.

At day 6, 3D spheroids (NT: non-treated control) and those treated with a 5 ng/mL solution of TGF-β2 (TGFβ) in the absence and presence of 100 nM PGF2α (PG) or the EP2 agonist, omidenepag (OMD), were immunostained with specific antibodies against COL6 (green), DAPI (blue), and phalloidin (Phal, red). Representative immunolabeling by anti-COL6 is shown in panel A (Scale bar: 100 µm) and the intensities of staining are plotted in panel B. All experiments were performed in duplicate using fresh preparations consisting of 10 spheroids each. Data presented (total *n* = 20 different 3D spheroids’ images) are the arithmetic mean ± standard error of the mean (SEM), and statistical differences were determined by ANOVA followed by a Tukey’s multiple comparison test. ** *p* < 0.01.

At day 6, 2D- and 3D-cultured HTM cells (NT: non-treated control) and those treated with a 5 ng/mL solution of TGF-β2 (TGFβ) in the absence and presence of 100 nM PGF2α (PG) or the EP2 agonist, omidenepag (OMD), were subjected to mRNA expression analysis of mRNA in *claudin11* (*Cldn11*) and *ZO1*. Analyses were performed in triplicate using fresh preparations (*n* = 12–15 3D spheroids each). Data presented are the arithmetic mean ± standard error of the mean (SEM), and statistical differences were determined by ANOVA followed by a Tukey’s multiple comparison test. * *p* < 0.05, ** *p* < 0.01, *** *p* < 0.005.

## 4. Discussion

As compared to conventional 2D cell cultures, 3D spheroid cell cultures have recently received considerable attention for use as suitable in vitro disease models including glaucoma using HTM cells [18,30,31,32]. However, there may also be some drawbacks to the use of these 3D culture techniques in terms of mimicking the physiological and pathological conditions of human TM due to the presence of unnecessary scaffolds or a collagen matrix. To avoid such potential problems, we recently developed a 3D cell drop culture method as an in vitro model for Graves’ orbitopathy [23], deepening upper eyelid sulcus (DUES) [25,27]. In our earlier pilot study, we also applied this method to replicate glaucomatous TM and found that TGF-β2 significantly induced the downsizing and stiffness of 3D HTM spheroids and that these effects were substantially inhibited by pan-ROCK inhibitors [17]. It is noteworthy that in the current study, we also observed several differences between TGF-β2-treated 2D and 3D HTM cells in terms of the mRNA expression of ECM molecules and their regulatory factors. As of this writing, our knowledge regarding the molecular mechanisms responsible for causing such diversities between 2D and 3D cells is very limited, despite the fact that the same cells are used. However, quite interestingly, the changes in the gene expressions of the ER stress-related factors tested were observed to be much less in the case of 3D HTM spheroids as compared with 2D HTM cells upon administering a 5 ng/mL solution of TGF-β2 (Appendix A). Therefore, these collective findings suggest that 3D spheroids may be a more stable environment as compared to the corresponding 2D cell cultures of HTM cells. In fact, taking advantage of the use of 3D spheroid cultures, we also reported that this method may be suitable for the screening of several anti-glaucoma drugs toward TGF-β2 and dexamethasone-treated 3D HTM spheroids as in vitro models for HTM with primary open-angle glaucoma and steroid-induced glaucoma [20,21,24].

In terms of the possible hypotensive effects of both the PGF2α-ags and EP2 agonists, it has been suggested that an increased uveoscleral outflow pathway through the activation of matrix metalloproteinases (MMP) and ECM remodeling is primarily involved [33,34,35,36,37]. However, it has also been suggested that both drugs may also affect the conventional TM outflow, the main route for AH drainage based on the presence of both FP and EP2 receptors on TM tissues [38,39,40,41]. FP and EP2 receptors are functionally characterized as contractile and relaxant receptors that are responsible for the contraction of smooth muscle cells [42,43,44,45,46] as well as Schlemm’s canal endothelial cells [46,47]. While, in contrast, it was reported that FP receptors mediated an enhancement in pulmonary and myocardial fibrosis [48,49], despite the fact that lung fibrosis had deteriorated upon the deletion of the EP2 receptor [50]. These collective observations suggest that the stimulation and inhibition of FP or EP2 receptors may modulate the structure and function of the TM in different ways. In the current study, we also found that the PGF2α and the EP2 agonist, OMD, had different effects on the barrier functions by TEER and FITC-dextran permeability analyses of the 2D HTM monolayers and the physical property analyses of the TGFβ2-induced 3D HTM spheroids. That is, as shown in Table 2, PGF2α induced a decrease in the TEER values of 2D HTM monolayers and an increase in the stiffness of the 3D HTM spheroids, but OMD had no effect on these parameters, although both PGF2α and OMD caused an increase in the size of the 3D HTM spheroids. Interestingly, the pan-ROCK inhibitors, ripasudil (Rip) also induced similar changes in the TEER values and 3D spheroid size but caused quite different effects on 3D spheroid stiffness as compared to PGF2α and OMD.

In terms of the fibrotic changes of TM cells, TGF-β2 activates cytoplasmic Smad2/3 leading to the up-regulation of the expression of various ECM molecules, such as FN and COL4, which induced an impediment to AH outflow through the TM, ultimately resulting in an elevation in IOP levels [51]. It was revealed that both the FP agonist, latanoprost, and the EP2 agonist, butaprost, inhibited TGF-β2–mediated collagen deposition [18]. In fact, a loss of collagen materials within the TM was also observed upon the treatment of cynomolgus monkeys with either latanoprost [34] or butaprost [33] for a period of one year. In addition, another FP agonist, fluprostenol, was reported to induce a decrease in the expression of COL4 and 6 that was induced by CTGF in cultured TM cells [52]. In the current study, we also found that TGF-β2 caused a significant up-regulation of the most of ECMs tested and that PGF2α or OMD induced a significant up-regulation of network-forming COL4 [53] and the down-regulation of beaded-filament-forming COL6 [53], FN, and αSMA. If 3D spheroid size and stiffness were mainly related with network-forming COL4 and fibril forming COL1, then the PG- or OMD-induced down-regulation of COL6, FN, and αSMA would result in a relative increase in COL1 content, which provides a possible explanation for why PG or OMD caused changes in both the size and stiffness of the 3D spheroids.

The findings presented in this study indicate that PGF2α and OMD significantly and differently modulated the TEER values and FITC-dextran permeability of TGF-β2-treated 2D HTM monolayers in addition to the physical properties of the TGF-β2-treated 3D HTM spheroids, as described above. It is well known that these barrier functions appear to exclusively depend on TJ properties [28,29]. In fact, in our current study, we also observed that PGF2α and OMD induced a significant up-regulation of Cldn11 and ZO1 in the 2D HTM monolayers, although no significant difference was observed between them despite the significant diverse effects toward TEER values (Figure 6). Therefore, these collective findings suggest that additional unknown mechanisms may well be involved, and the following study limitations that need to be discussed are as follows; (1) rationale mechanisms for causing diversity in the permeability of the 2D NTM monolayer between PGF2α and OMD which have not been fully elucidated, and (2) the relationship between such collective observations and the ease of aqueous outflow of the human TM for a better understanding of the roles of FP or EP2 agonists in the AH dynamics of HTM. Therefore, further investigation will be required to obtain additional insights into the molecular mechanisms causing diverse effects by PGF2α and OMD for future therapeutic strategies for the treatment of glaucoma using PGF2α-ags and EP2 agonist. 

## Figures and Tables

**Figure 1 jcm-11-01652-f001:**
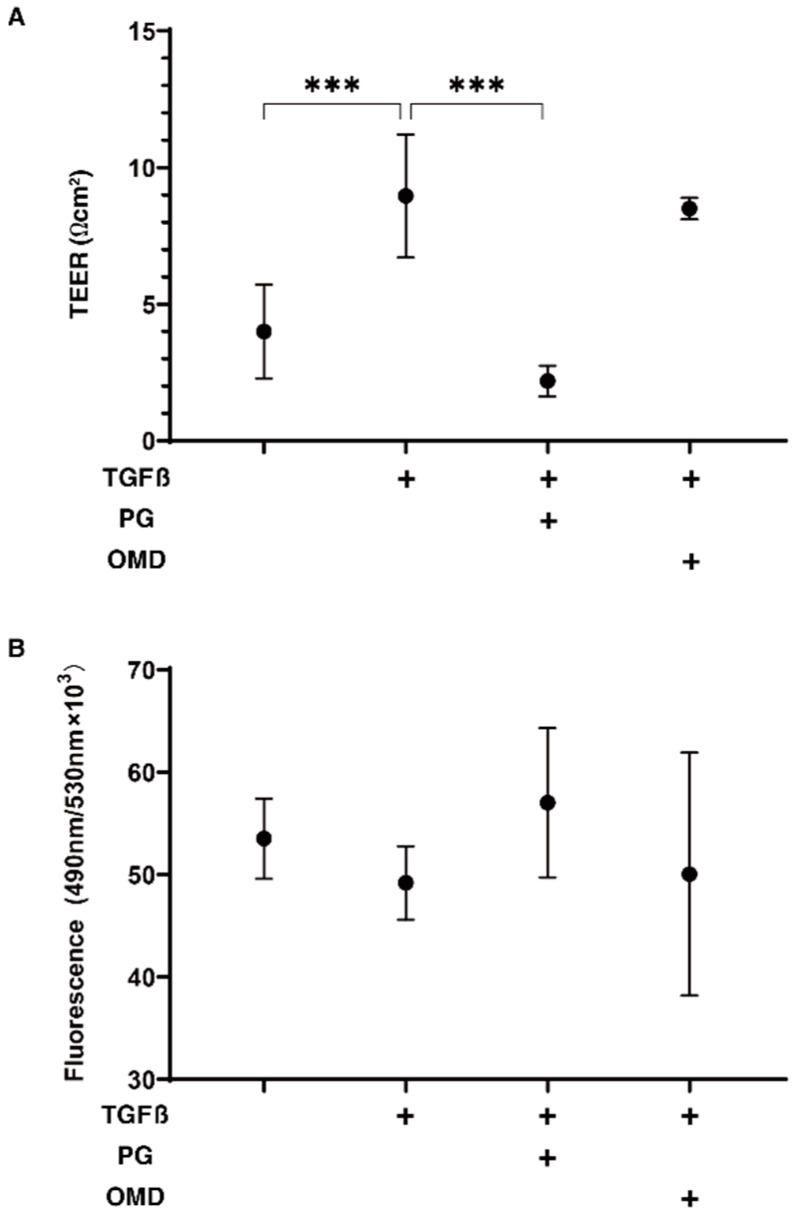
Effects of PGF2α and the EP2 agonist, omidenepag (OMD), on barrier functions of TGFβ2-treated 2D HTM monolayers by transendothelial electrical resistance (TEER) (**A**) and FITC-dextran permeability (**B**). To compare the effects of 100 nM PGF2α (PG) and omidenepag (OMD) on the barrier functions of 5 ng/mL TGF-β2 (TGFβ)-treated 2D HTM monolayers, TEER values (Ωcm^2^) and FITC-dextran permeability were measured in addition to the TGFβ untreated control, and these values were plotted in panel A and B, respectively. “+” is reagents addition. All experiments were performed using fresh preparations (*n* = 3–4). All data are presented as the arithmetic mean ± standard error of the mean (SEM). *** *p* < 0.005 (ANOVA followed by a Tukey’s multiple comparison test).

**Figure 2 jcm-11-01652-f002:**
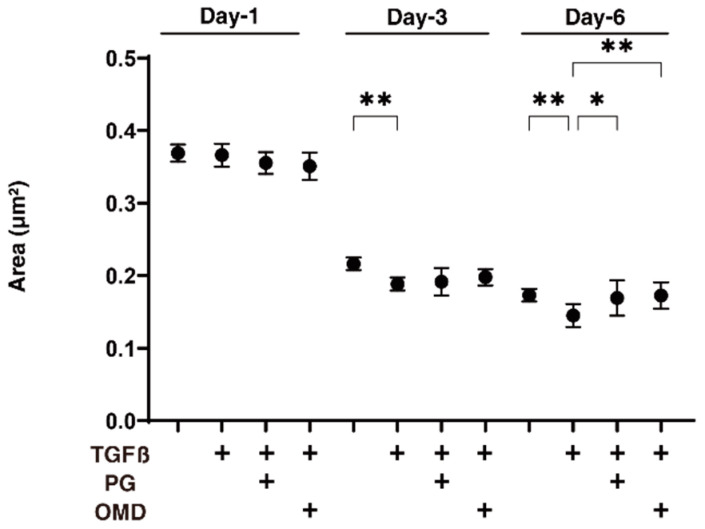
Effects of PGF2α and the EP2 agonist, omidenepag (OMD), on sizes of TGFβ2-treated 3D HTM spheroids at days 1, 3, or 6.To compare the effects of 100 nM PGF2α (PG) and omidenepag (OMD) on the mean sizes of the 5 ng/mL TGF-β2 (TGFβ) HTM 3D spheroids, measurements were made on days 1, 3, or 6 in addition to TGFβ untreated controls, and plotted (*n* = 10–12 3D spheroids for each experimental condition). “+” is reagents addition. Data that are presented are the arithmetic mean ± standard error of the mean (SEM), and statistical differences were determined by ANOVA followed by a Tukey’s multiple comparison test. * *p* < 0.05, ** *p* < 0.01.

**Figure 3 jcm-11-01652-f003:**
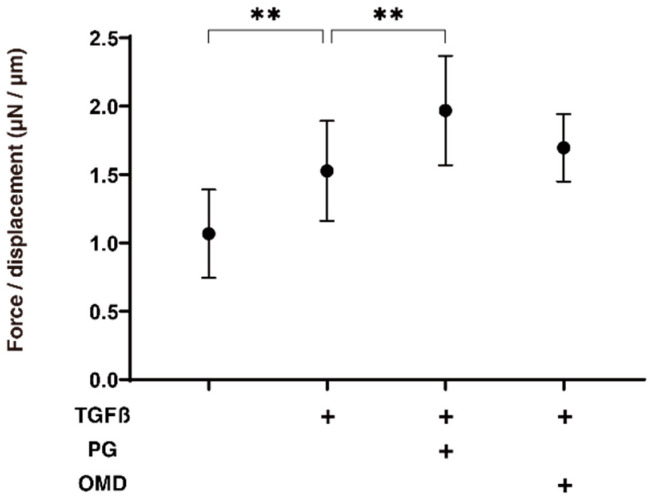
Effects of PGF2α and the EP2 agonist, omidenepag (OMD), on stiffness of TGFβ2-treated 3D HTM spheroids.To compare the effects of 100 nM PGF2α (PG) and omidenepag (OMD) on the stiffness of the 5 ng/mL TGF-β2 (TGFβ) HTM 3D spheroids, those at day 6 were subjected to a micro-squeezer analysis in addition to TGFβ untreated controls. The required force (μN) was measured and force/displacement (μN/μm) were plotted (*n* = 10–12 3D spheroids in each experimental condition). “+” is reagents addition. Data presented are the arithmetic mean ± standard error of the mean (SEM), and statistical differences were determined by ANOVA followed by a Tukey’s multiple comparison test. ** *p* < 0.01.

**Figure 4 jcm-11-01652-f004:**
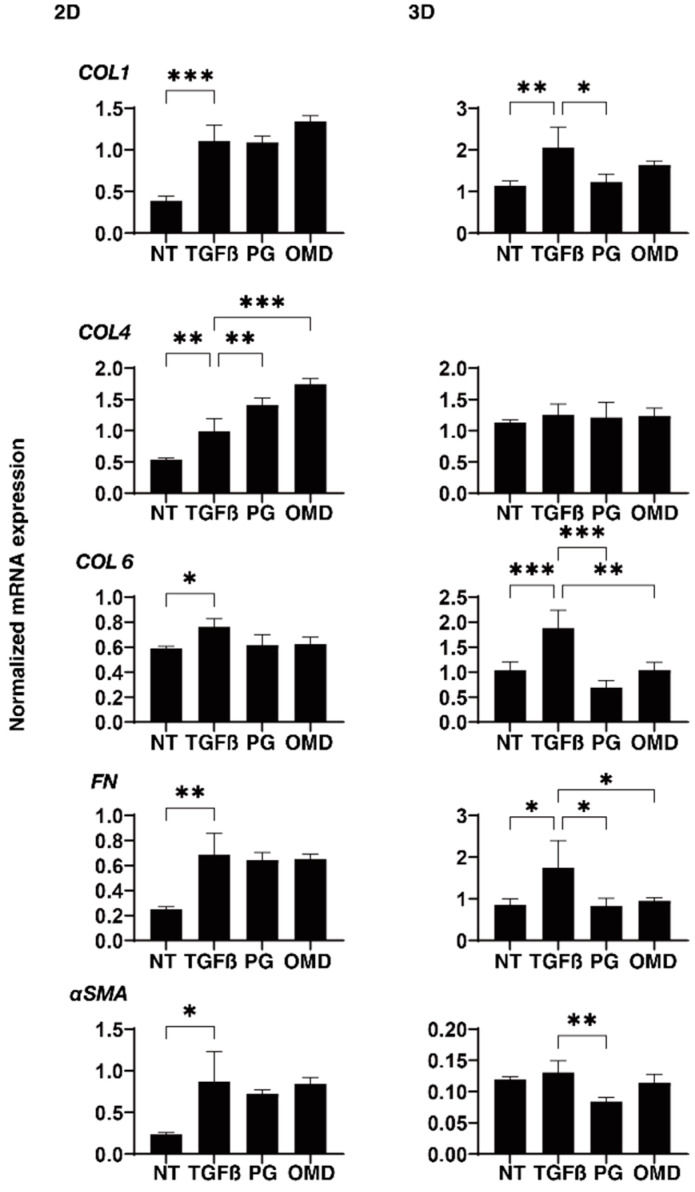
Effects of PGF2α and the EP2 agonist, omidenepag (OMD), on the mRNA expression of ECM proteins in 2D- and 3D-cultured HTM cells. At Day 6, HTM 2D cells and 3D spheroids (NT: non-treated control) and those treated with a 5 ng/mL solution of TGF-β2 (TGFβ) in the absence and presence of 100 nM PGF2α (PG) or the EP2 agonist, omidenepag (OMD) were subjected to qPCR analysis to estimate the expression of mRNA in ECMs (COL1, COL4, COL6, FN and αSMA). Analyses were performed in triplicate using fresh preparations (*n* = 12–15 3D spheroids each). “+” is reagents addition. Data presented are the arithmetic mean ± standard error of the mean (SEM), and statistical differences were determined by ANOVA followed by a Tukey’s multiple comparison test. * *p* < 0.05, ** *p* < 0.01, *** *p* < 0.005.

**Figure 5 jcm-11-01652-f005:**
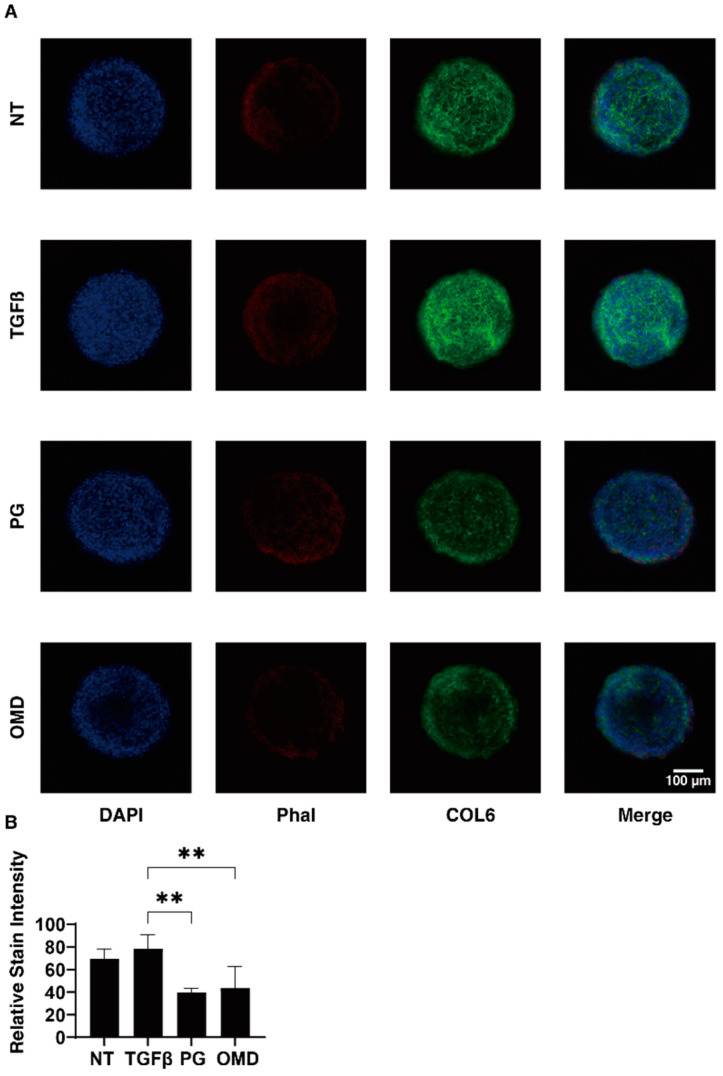
Immunofluorescence images of the expressed COL6 in 3D HTM spheroids. At Day 6, 3D spheroids (NT: non-treated control) and those treated with a 5 ng/mL solution of TGF-β2 (TGFβ) in the absence and presence of 100 nM PGF2α (PG) or the EP2 agonist, omidenepag (OMD) were immunostained with specific antibodies against COL6 (green), DAPI (blue) and phalloidin (Phal, red). Representative immunolabeling by anti-COL6 are shown in panel (**A**) (Scale bar: 100 µm) and the intensities of staining are plotted in panel (**B**). All experiments were performed in duplicate using fresh preparations consisting of 10 spheroids each. Data presented (total *n* = 20 different 3D spheroids’ images) are the arithmetic mean ± standard error of the mean (SEM), and statistical differences were determined by ANOVA followed by a Tukey’s multiple comparison test. ** *p* < 0.01.

**Figure 6 jcm-11-01652-f006:**
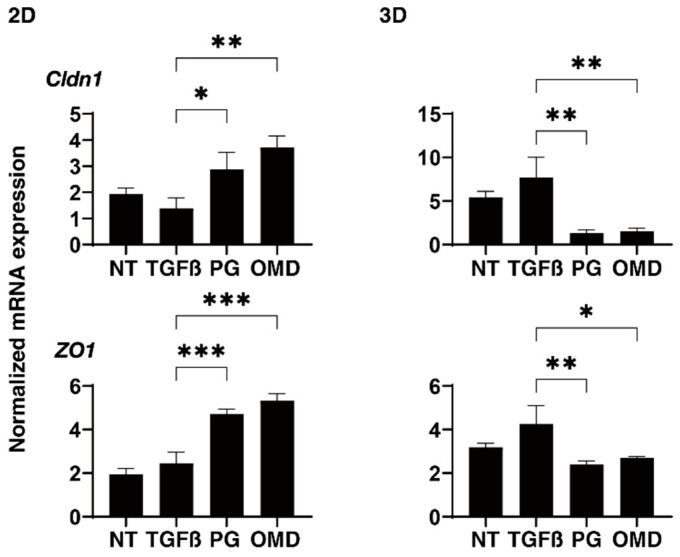
Effects of PGF2α and the EP2 agonist, omidenepag (OMD), on the mRNA expressions of tight junction related proteins in 2D- and 3D-cultured HTM cells. At Day 6, 2D and 3D cultured HTM cells (NT: non-treated control) and those treated with a 5 ng/mL solution of TGF-β2 (TGFβ) in the absence and presence of 100 nM PGF2α (PG) or the EP2 agonist, omidenepag (OMD) were subjected to mRNA expression analysis of mRNA in claudin11 (Cldn11) and ZO1. Analyses were performed in triplicate using fresh preparations (*n* = 12–15 3D spheroids each). Data presented are the arithmetic mean ± standard error of the mean (SEM), and statistical differences were determined by ANOVA followed by a Tukey’s multiple comparison test. * *p* < 0.05, ** *p* < 0.01, *** *p* < 0.005.

**Table 1 jcm-11-01652-t001:** Summary of mRNA expressions of ECM proteins, tight junction-related molecules, TIMPS, MMPs, CRGF, and ER stress-related factors.

	2D			3D		
	TGF	PG	OMD	TGF	PG	OMD
COL1	↑↑			↑↑	↓	
COL4	↑↑	↓↓	↓↓			
COL6	↑			↑↑	↓↓	↓
FN	↑↑			↑	↓	↓
αSMA	↑				↓↓	
Cldn11		↑	↑↑		↓↓	↓↓
ZO1		↑↑	↑↑		↓↓	↓
TIMP 1		↓			↓↓	
TIMP 2	↑					
TIMP 3				↑↑	↓↓	↓↓
TIMP 4				↑		
MMP 2	↑			↑↑		
MMP 9	↑			↑↑	↓	
MMP 14	↑			↑↑		
CTGF	↑↑	↑	↑	↑		
GRP 78		↓	↓↓			
GRP 94	↑↑	↓↓	↓↓	↓↓		
XBP	↑↑	↓↓	↓↓	↓		
sXBP	↑↑	↓↓	↓↓	↓↓		
CHOP	↑↑	↓↓	↓↓		↓	

Statistically significant differences between TGF untreated control and TGF, or TGF and PG or OMD are designated as follows; ↑: significant increase (*p* < 0.05), ↑↑: significant increase (*p* < 0.01), ↓: significant decrease (*p* < 0.05), ↓↓: significant decrease (*p* < 0.01).

**Table 2 jcm-11-01652-t002:** Comparison of the drug-induced effects on the TEER values of the 2D HTM monolayers, and physical properties, size and stiffness, and mRNA expression of ECM molecules with 3D HTM spheroids.

	PGF2α	OMD	Rip*
2D monolayer			
TEER	↓	(−)	↓
3D spheroid			
size	↑	↑	↑
stiffness	↑	(−)	↓
ECM expression			
2D	*COL4* ↑	*COL4* ↑	(N.D.)
3D			*COL1* ↓
			*COL4* ↓
	*COL6* ↓	*COL6* ↓	
	*FN* ↓	*FN* ↓	
	*αSMA* ↓		

PGF2α; prostaglandin F2α, OMD; omidenepag, Rip; ripasudil, TEER; Transendothelial electron resistance, *COL1*; collagen 1, *COL4*; collagen 4, *COL6*; collagen 6, *FN*; fibronectin, α*SMA*; α smooth muscle actin, ↑: significant increase, ↓: significant decrease, (–): not significant change. N.D.; not determined. Data related to Rip* were recruited from our previous study [17].

## Data Availability

The data that support the findings of this study are available from the corresponding author upon reasonable request.

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
