# Peer review of "Comparison of the Drug-Induced Efficacies between Omidenepag Isopropyl, an EP2 Agonist and PGF2α toward TGF-β2-Modulated Human Trabecular Meshwork (HTM) Cells"

_jcm, 2022, doi:10.3390/jcm11061652_

Round 1
Reviewer 1 Report
The manuscript has improved since last time. Despite being grammatically correct with only few spelling mistakes still present, the reading remains somewhat very difficult. Main suggestion to the authors would be to make the entire manuscript more concise and focused, please consider reducing the number of figures and overall data presented - some may well be part of a companion manuscript. As it is, data still appear somewhat confusing and difficult to follow.
Author Response
Dear Editor,
Thank you very much again for the constructive comments concerning our manuscript; " Comparison of the drug induced effects of a selective EP2 agonist, Omidenepag isopropyl, with PGF2α, on TGF-β2-treated human trabecular meshwork (HTM) cells “. We carefully examined all of the comments from the Reviewer and have made a series of specific changes in our paper that were suggested by the reviewer. The changes that were made are highlighted and our responses to the comments are listed below;
During the course of preparation of the revision of the manuscript, the author contribution was significantly changed. Therefore, we changed theorder of authors (Soma Suzuki, Masato Furuhashi, Yuri Tsugeno, Araya Umetsu, Yosuke Ida, Fumihito Hikage, Hiroshi Ohguro, Megumi Watanabe) and the corresponding author is now listed as Hiroshi Ohguro. In addition, Masato Furuhashi made a substantial contribution to this study as did Soma Suzuki. Therefore, in the revised version of our paper, both authors contributed equally. In addition, Megumi Watanabe also contributed substantially in preparing the revised form of the manuscript. Therefore, I felt that Megumi Watanabe should now be listed as the corresponding author.
Editorial comment;
We have attached a report that indicates the similarity between your manuscript and other published manuscripts (63%), which is not allowed by our company.
Please revise the highly similarity sentences according to the report (even seven continuous words in one sentence).
Answer; As suggested, entire manuscript was edited to avoid the highly similarity sentences as pointed out.
Reviewer1
The manuscript has improved since the last time. Despite being grammatically correct with only few spelling mistakes still present, the reading remains somewhat very difficult. Main suggestion to the authors would be to make the entire manuscript more concise and focused, please consider reducing the number of figures and overall data presented - some may well be part of a companion manuscript. As it is, data still appear somewhat confusing and difficult to follow.
Answer; Thank you for this comment. As suggested, to make the manuscript more concise and focused, we reduced the numbers of figures as follows; 1) since immunofluorescence images of ECM proteins of the 2D cultured HTM cells were quite similar in terms of their mRNA expressions, Fig. 5 was moved to supplemental Fig.2, 2) important data related to the mRNA expression of tight junction (TJ) molecules, claudin11 and ZO1 are now included, as requested by reviewer 2, 3) those of TIMPs, MMPs, CTGF and ER-stress related factors were removed to supplemental figures, 4) instead, a new Table 1 summarizes the qPCR data of TJ molecules, TIMPs, MMPs, CTGF and ER-stress related factors was included, and 4) Discussion was rewritten in a more focused and compact shape. In terms of English correction, I asked Dr. Milton, an excellent native English speaking scientist, to edit current manuscript and he has done so.

Reviewer 2 Report
This manuscript by Suzuki et al investigated the effects of, Omidenepag isopropyl (OMD) , a selective EP2 agonist, and PGF2α on TGF-β2-treated human trabecular meshwork cells. I have several comments and suggestions for the authors.
- Please list the sources of the antibodies as well as the specificity of these antibodies employed if these antibodies have been tested and characterized previously. Otherwise, the authors need to make sure the specificity of these antibodies.
- It appears the immunofluorescence labeling on Figures 5 and 6 are not of high quality, because the authors used a 20X objective lens under a confocal microscopy. I strongly suggest the authors to use at least 40X oil objective lens. In addition, please add DAPI on Figure 5.
- It appears that some of the PCR primers also accompany with a probe sequence, but many primers don’t accompany with a probe sequence (From pages P20-P21), does that means different primers using different protocol for running a real time RT-PCR? In addition, please list all the predicted PCR product length.
- The authors need to give a deep discussion regarding the mechanism that OMD or PGF2α can regulate TEER, because the TEER is related to tight junction formed barrier functions, that means these drugs may regulate tight junctions and then change TEERs. Otherwise, this manuscript is most descriptive work.
- Minor question, on pages 4 and 14, some alpha symbol letter for SMA have been changed.
Thank you for the invitation!
Author Response
Dear Editor,
Thank you very much again for the constructive comments concerning our manuscript; " Comparison of the drug induced effects of a selective EP2 agonist, Omidenepag isopropyl, with PGF2α, on TGF-β2-treated human trabecular meshwork (HTM) cells “. We carefully examined all of the comments from the Reviewer and have made a series of specific changes in our paper that were suggested by the reviewer. The changes that were made are highlighted and our responses to the comments are listed below;
During the course of preparation of the revision of the manuscript, the author contribution was significantly changed. Therefore, we changed theorder of authors (Soma Suzuki, Masato Furuhashi, Yuri Tsugeno, Araya Umetsu, Yosuke Ida, Fumihito Hikage, Hiroshi Ohguro, Megumi Watanabe) and the corresponding author is now listed as Hiroshi Ohguro. In addition, Masato Furuhashi made a substantial contribution to this study as did Soma Suzuki. Therefore, in the revised version of our paper, both authors contributed equally. In addition, Megumi Watanabe also contributed substantially in preparing the revised form of the manuscript. Therefore, I felt that Megumi Watanabe should now be listed as the corresponding author.
Editorial comment;
We have attached a report that indicates the similarity between your manuscript and other published manuscripts (63%), which is not allowed by our company.
Please revise the highly similarity sentences according to the report (even seven continuous words in one sentence).
Answer; As suggested, entire manuscript was edited to avoid the highly similarity sentences as pointed out.
Reviewer2
- Please list the sources of the antibodies as well as the specificity of these antibodies employed if these antibodies have been tested and characterized previously. Otherwise, the authors need to make sure the specificity of these antibodies.
Answer; As suggested, the sources of the antibodies are now listed as well as the specificity of these antibodies in supplemental Table 1.
- It appears the immunofluorescence labeling on Figures 5 and 6 are not of high quality, because the authors used a 20X objective lens under a confocal microscopy. I strongly suggest the authors to use at least 40X oil objective lens. In addition, please add DAPI on Figure 5.
Answer; As suggested, immunofluorescence images were changed to higher magnified ones with DAPI in both Figs. However, as the request to reduce the numbers of figures by Reviewer 1, 2D images was moved to supplemental figure since immunofluorescence images of ECM proteins of the 2D cultured HTM cells were almost similar with their mRNA expressions.
- It appears that some of the PCR primers also accompany with a probe sequence, but many primers don’t accompany with a probe sequence (From pages P20-P21), does that means different primers using different protocol for running a real time RT-PCR? In addition, please list all the predicted PCR product length.
Answer; In terms of qPCR analysis, we used different primer sources, Tagman (no probe is required) and SYBR Green (probe is required). As pointed out, supplemental Table 2 lists the source of the PCR primers.
- The authors need to give a deep discussion regarding the mechanism that OMD or PGF2α can regulate TEER, because the TEER is related to tight junction formed barrier functions, that means these drugs may regulate tight junctions and then change TEERs. Otherwise, this manuscript is most descriptive work.
Answer; Thank you and I completely agree with this comment. As suggested, additional data related to the qPCR analysis of tight junction molecules, claudin11 and ZO1 are now included as new Fig. 7 and those of TIMPs, MMPs, CTGF and ER-stress related factors were removed to supplemental figures. Instead, a new Table 1 was prepared that summarizes the qPCR data of TJ molecules, TIMPs, MMPs, CTGF and ER-stress related factors and is now was included in the paper. In addition, a discussion of the relationship between TJ molecule expression and TEER changes was included.
- Minor question, on pages 4 and 14, some alpha symbol letter for SMA have been changed.
Answer; As pointed out, the erroneous symbols are now corrected.

Round 2
Reviewer 2 Report
I have checked the revised manuscript carefully. There are still some issues need to be addressed.
The authors’ answer to my previous question 3 is not right regarding the probe usage in real time qPCR utilizing SYBR or TaqMan based methods. From my understanding, you must use probes for TaqMan based technique, while no probe is needed for SYBR based qPCR analysis.
In addition, it appears the immunofluorescence labeling quality is not of high quality. I add one of my immunofluorescent labeling of Collagen VI (in green color) in primary cultured TM cells using an Olympus confocal microscope with 40 X oil lens as an example.
“3.It appears that some of the PCR primers also accompany with a probe sequence, but many primers don’t accompany with a probe sequence (From pages P20-P21), does that mean different primers using different protocol for running a real time RT-PCR? In addition, please list all the predicted PCR product length.
Answer; In terms of qPCR analysis, we used different primer sources, Tagman (no probe is required) and SYBR Green (probe is required). As pointed out, supplemental Table 2 lists the source of the PCR primers”.

Author Response
Dear Editor,
Thank you very much again for the constructive comments concerning our manuscript; " Comparison of the drug induced effects of a selective EP2 agonist, Omidenepag isopropyl, with PGF2α, on TGF-β2-treated human trabecular meshwork (HTM) cells “. We carefully examined all of the comments from the Reviewer and have made a series of specific changes in our paper that were suggested by the reviewer. The changes that were made are highlighted and our responses to the comments are listed below;
Reviewer 2 comments
I have checked the revised manuscript carefully. There are still some issues need to be addressed.
The authors’ answer to my previous question 3 is not right regarding the probe usage in real time qPCR utilizing SYBR or TaqMan based methods. From my understanding, you must use probes for TaqMan based technique, while no probe is needed for SYBR based qPCR analysis.
Answer; Thank you for this comment and suggestion. In terms of SYBR based qPCR analysis for some genes, those were already established method quite solidly in our and other Lab. Thus, we would not dare switch to TaqMan based technique. Alternatively, others were newly started for analysis, and therefore used TaqMan based technique.
In addition, it appears the immunofluorescence labeling quality is not of high quality. I add one of my immunofluorescent labeling of Collagen VI (in green color) in primary cultured TM cells using an Olympus confocal microscope with 40 X oil lens as an example.
Answer; Thank you for demonstrating such an excellent the immunofluorescence labeling. I was so impressed. As suggested, we tried to the immunofluorescence labeling again using confocal microscope with 40 X oil lens and thus corresponding pictures were replaced within supplemental Fig. 1.
“3.It appears that some of the PCR primers also accompany with a probe sequence, but many primers don’t accompany with a probe sequence (From pages P20-P21), does that mean different primers using different protocol for running a real time RT-PCR? In addition, please list all the predicted PCR product length.
Answer; Thank you for this comment. As suggested, I prepared new supplemental table 1B describing protocols for running a real time RT-PCR based on TaqMan based and SYBR based analysis, and list all the predicted PCR product length were included within supplemental Table 1A.

This manuscript is a resubmission of an earlier submission. The following is a list of the peer review reports and author responses from that submission.
Round 1
Reviewer 1 Report
The manuscript by Watanabe et al., attempt to characterize a relatively new model of glaucoma employing HTM cells; furthermore they address the potential effects of PGF2alpha and Omidenepag in this model. The approach is somewhat interesting, however, several issues can be noted:
- Language requires major editing. Grammatical mistakes and inconclusive statements are present throughout the manuscript - this is just unacceptable as it makes the reading really challenging.
- The specific rationale why two compounds acting on uveoscleral pathway are tested here is neither clear nor discussed. This model should resemble conventional outflow pathway which is a minor targets for both compounds.
- In many occasions, statistics seem inappropriate as it revels highly significant values even if changes are rather modest and errors overlap i.e. fig 2 and fig.3.
- Some figure legends (see fig. 3) requires editing as it report on groups not present in the actual plot.
- Rationale for compounds concentration is not expressed; actually, the concentration used is not stated clearly. Concentration-response curves are missing making very difficult any inference on the meaning and the relevance of the data presented.
- The model is often referred to as an "in vivo model" raising a fundamental question on as to whether the authors really understand their own model. This model, if anything, is an in vitro model - suggestion is to first understand the model and its relevance to then start addressing the effects of compounds.
- Discussion is confusing and misleading - very difficult to understand the points they are trying to make. Furthermore negative data (fig.4-5-6-7) are not discussed. Differences in findings between the two models employed is not discussed.
Author Response
Dear Editor,
Thank you very much for the constructive comments concerning our manuscript; " Comparison of the drug induced effects of a selective EP2 ago-nist, Omidenepag isopropyl, with PGF2α, on TGF-β2-treated human trabecular meshwork (HTM) cells “. We carefully examined all of the comments from the Reviewer and have made a series of specific changes to our manuscript as follows;
During the course of preparation of 1st draft of the manuscript and this correction, author contribution was significantly changed. Therefore, we want change order of authors (Megumi Watanabe, Masato Furuhashi, Yosuke Ida, Fumihito Hikage, Hiroshi Ohguro,) and corresponding author is Hiroshi Ohguro.
Reviewer1
The manuscript by Watanabe et al., attempt to characterize a relatively new model of glaucoma employing HTM cells; furthermore they address the potential effects of PGF2alpha and Omidenepag in this model. The approach is somewhat interesting, however, several issues can be noted:
- Language requires major editing. Grammatical mistakes and inconclusive statements are present throughout the manuscript - this is just unacceptable as it makes the reading really challenging.
Answer; Thank you for this comment. In terms of English editing, we asked to a native speaking scientist, Dr. Milton Feather to improve this manuscript. Certification from Dr. Feather is attached. He shortened many of the long sentences in the manuscript.
- The specific rationale why two compounds acting on uveoscleral pathway are tested here is neither clear nor discussed. This model should resemble conventional outflow pathway which is a minor targets for both compounds.
Answer; Thank you for this comment regarding the scientific rationale for why two compounds, PGF2a and OMD, acting on uveoscleral pathway were tested. I agree with you that both compounds belong the same category of PG-related molecules and also that both act toward aqueous outflow by the uveoscleral pathway. However, our recent studies demonstrated that both compounds induced quite different effects toward adipogenesis of the 3T3-L1 mouse preadipocytes (Sci Rep. 2020 Sep 29;10(1):16018. doi: 10.1038/s41598-020-72538-x, Int J Mol Sci. 2021 Apr 28;22(9):4648. doi: 10.3390/ijms22094648, Prostaglandins Leukot Essent Fatty Acids. 2021 Aug;171:102315. doi: 10.1016/j.plefa.2021.102315. Epub 2021 Jul 3) as well as human orbital fibroblasts (HOF, Transl Vis Sci Technol. 2021 Apr 1;10(4):6. doi: 10.1167/tvst.10.4.6) and Graves’ orbitopathy related HOF (Exp Eye Res. 2021 Apr;205:108489. doi: 10.1016/j.exer.2021.108489. Epub 2021 Feb 12). Furthermore, diverse effects exerted by these compounds were also recognized in the case of dexamethasone treated HTM cells as a steroid induced glaucoma model (Biomedicines. 2021 Jul 31;9(8):930. doi: 10.3390/biomedicines9080930). Therefore, this information is included in the 2nd paragraph of the introduction; “In addition to prostaglandin (PG) F2α agonists (PGF2α-ags) as first-line drugs for the treatment of GON 10-12, a non-prostaglandin, a prostanoid EP2 agonist Omidenepag isopropyl (OMDI), which is converted into the active form (Omidenepag, OMD) by hydrolysis after its administration has recently been made available for the treatment of patients with OH and POAG 13-14. It is generally thought that the hypotensive mechanisms by both PGF2α-ags and the EP2 agonist, OMD, function to increase the ease of uveoscleral outflow, but not the ease of conventional outflow 15. However, an in vivo study using OH monkeys reported that the dynamics of the pharmacokinetics of OMD in the AH are quite different from those of PGF2α-ags 16. Therefore, although both PGF2α-ags and OMD similarly affect AH outflow by the uveo-scleral root and are categorized as the same members of PG-related compounds, OMD may cause different effects from PGF2α-ags toward human TM, especially glaucomatous human TM. In fact, the effects of these drugs on the ease of conventional outflow have not been extensively characterized at this time, since an appropriate experimental model for evaluating the drug efficacies on conventional versus uveoscleral roots has not been available. To address this difficulty, we recently developed an in vivo model that replicates the human TM (HTM) using a unique 3D drop culture method 17, in which no assistance such as a collagen-gel or a scaffold that is usually used in the 3D culture methods 18 were required. Furthermore, when using this model for steroid-induced glaucoma for evaluat-ing PGF2α-ags and OMD, significant diverse effects were detected in the dexamethasone treated HTM cells 19, in addition to adipogenesis of the 3T3-L1 mouse preadipocytes 20-21 as well as human orbital fibroblasts (HOF) 22 and Graves’ orbitopathy related HOF23. “.
- In many occasions, statistics seem inappropriate as it revels highly significant values even if changes are rather modest and errors overlap i.e. fig 2 and fig.3.
Answer; Thank you for this comment. In terms of the statistical significance between each two pairs, we used the t-test and confirmed that our statistical significances were correct; Fig. 2; Day3 NT vs TGF (p<0.001), Day 6 NT vs TGF (p<0.001), TFG vs TGF/PG (p=0.02), TFG vs TGF/OMD (p=0.002), Fig. 3; NT vs TGF (p=0.003), TGF vs TGF/PG (p=0.004).
- Some figure legends (see fig. 3) requires editing as it report on groups not present in the actual plot.
Answer; As suggested, legend of Fig. 3 was changed; “Figure 3. Physical solidity of 3D HTM spheroids. At Day 6, the physical solidity by a micro-squeezer (μN/μm force/displacement) of HTM 3D spheroids and those treated by 5 ng/ml TGF-β2 (TGFβ) in the absence and presence of 100 nM PGF2α (PG) or EP2 agonist, omidenepag (OMD) were plotted. All experiments were performed in using fresh preparations (n=10-12 3D spheroids). Data are presented as the arithmetic mean ± standard error of the mean (SEM). **P<0.01 (ANOVA followed by a Tukey’s multiple comparison test).”
- Rationale for compounds concentration is not expressed; actually, the concentration used is not stated clearly. Concentration-response curves are missing making very difficult any inference on the meaning and the relevance of the data presented. Answer; Thank you for this question. In terms of this issue, as the scientific basis of the drug concentrations used in the current study, we followed a previous study that involved the testing of the dose dependency of PGF2a toward 2D cultured HTM cells (EC50=120nM) (Human trabecular meshwork cell responses induced by bimatoprost, travoprost, unoprostone, and other FP prostaglandin receptor agonist analogues. Invest Ophthalmol Vis Sci. 2003 Feb;44(2):715-21. doi: 10.1167/iovs.02-0323.) This information is included in the Methods section; “Human trabecular meshwork (HTM) cells
All experiments involving human tissue/cells were performed in compliance with the tenets of the Declaration of Helsinki, and all experimental protocols were approved by internal review board of Sapporo Medical University. Commercially available certified immortalized HTM cells transfected with an original defective mutant of the SV40 virus (Applied Biological Materials Inc., Richmond Canada) were used in the present study. To ensure that these HTM cells were truly TM cells, the DEX induced up-regulation of the mRNA expression of myocilin and extra domain A (EDA) fibronectin was confirmed among the criteria described in the consensus recommendations for TM cells reported by Keller et al. 19. In terms of the scientific basis of the drug concentrations used in the current study, the concentrations were consistent with a previous study the involved testing the dose de-pendency of PGF2a toward 2D cultured HTM cells (EC50=120nM) 25.”.
- The model is often referred to as an "in vivo model" raising a fundamental question on as to whether the authors really understand their own model. This model, if anything, is an in vitro model - suggestion is to first understand the model and its relevance to then start addressing the effects of compounds.
Answer; Thank you for this question. In terms of the current 2D and 3D HTM cultured models, we extensively characterized these models and reported on this in our recent study (Establishment of appropriate glaucoma models using dexamethasone or TGFβ2 treated three-dimension (3D) cultured human trabecular meshwork (HTM) cells. Sci Rep. 2021 Sep 29;11(1):19369. Doi: 10.1038/s41598-021-98766-3). Consequently, we believed that the 2D cultured HTM monolayer represents single sheet of human TM, and the 3D cultured HTM spheroid represent a multiple sheet structured human TM because DAPI staining of HTM nuclei were aligned concentrically within the soheroid. Based upon these observations, we assumed that our established 3D HTM spheroid well replicate human TM structure and thus this model may be an in vivo model rather than an in vitro model. Therefore, this information is included in the first paragraph of result; “In our recent study, we established a clinically relevant model replicating the human TM structure that is applicable for use in the field of glaucomatous research. 2D and 3D cultured HTM cells were prepared and were extensively characterized 28. The results indicated that the 2D cultured HTM monolayer represents a single sheet of human TM, and the 3D cultured HTM spheroids represent multiple sheet structured human TM. This conclusion was based on nuclear staining by DAPI indicating that HTM nuclei were aligned concentrically within the spheroid. Based upon these observations, we assumed that our established 3D HTM spheroid replicated the human TM structure reasonably well. We then used model as an in vivo model rather than an in vitro model.”.
- Discussion is confusing and misleading - very difficult to understand the points they are trying to make. Furthermore negative data (fig.4-5-6-7) are not discussed. Differences in findings between the two models employed is not discussed.
Answer; As suggested, to make the discussion more simpler, the 3rd paragraph of the discussion section was rewritten; “To elucidate the underlying mechanisms responsible for causing such diverse effects between PGF2α and OMD as above, we evaluated the gene expressions of several molecules that might possibly be determinants of the structures of the 2D and 3D cultured HTM cells. These studies included ECMs, ECM regulatory factors, TIMPs and MMPs, and ER-stress related factors (Figs. 4-7). However, as summarized in Table 1, the mRNA ex-pressions of these factors within the 2D and 3D cultured HTM cells were not significantly different between PGF2α and OMD. However, the current study has the following limitation that need to be taken into consideration; 1st, the relationship between such collective observations and the ease of aqueous outflow of the human TM remains to be elucidated. 2nd, we observed a consistent TGFb2-induced increase in the TEER values of the 2D HTM monolayer.17, 28 In contrast, however, the reverse TGF-b2-induced effects in the TEER, i.e., a decrease, has been reported in several studies using other sources of epithelial tissue 51-53. We currently have no explanation for why such TGF-b2-induced effects were different between HTM cells and other epithelial tissue. Nakamura et al. also studied effect of TGF and other factors on 2D HTM monolayers using TEER measurement. The conclusions reached in that study indicated that the TEER values were not statistically changed by a 24-hour exposure to TGFb1 or 254. In contrast to this, our observations consistently showed a TGFb2 induced increase in TEER values, in which the models were exposed to TGFb2 during Day1 through Day617, 28. These collective results indicate that, at a mini-mum, TGFb2 caused no change or increase in the TEER values, but not a decrease, and the longer exposure of TGFb2 may be related to differences between the study by Nakamura et al. 54 and ours. Since it is also well known that elevated levels of TGF-b2 were detected within AH obtained from patients with glaucoma. 55 It is possible that such continuous effects caused by TGF-b2 exposure may likely cause similar effects to those observed in the current study even though TGF-b2 is metabolized faster. In addition, the action of TGF-b2 on HTM cell function is currently unknown. However, since it appears that if PGF2α or OMD can actually alleviate a TGF-b2 induced increase in TEER, this rationally suggests possibility that PGF2α or OMD has some effects on TGF-b2 receptors. Thus, for a better understanding of the roles of FP or EP2 agonists within the AH dynamics of HTM, further studies to clarify above issues will be required to obtain additional insights into future therapeutic strategies for the treatment of glaucoma using PGF2α-ags and EP2 agonist.”.

Reviewer 2 Report
The article is well written, it is very complicate to understand results expecially from page 6 to page 7. Anyway I'm not qualified to judge this job. Conclusions are appropriate, maybe you can try to condense page 6 and 7 in order to have a much more easy reading article.
Author Response
- The article is well written, it is very complicate to understand results expecially from page 6 to page 7. Anyway I'm not qualified to judge this job. Conclusions are appropriate, maybe you can try to condense page 6 and 7 in order to have a much more easy reading article.
Answer; As suggested, to make the discussion section simpler, the 3rd paragraph of discussion was rewritten; “To elucidate the underlying mechanisms responsible for causing such diverse effects between PGF2α and OMD as above, we evaluated the gene expressions of several molecules that might possibly be determinants of the structures of the 2D and 3D cultured HTM cells. These studies included ECMs, ECM regulatory factors, TIMPs and MMPs, and ER-stress related factors (Figs. 4-7). However, as summarized in Table 1, the mRNA ex-pressions of these factors within the 2D and 3D cultured HTM cells were not significantly different between PGF2α and OMD. However, the current study has the following limitation that need to be taken into consideration; 1st, the relationship between such collective observations and the ease of aqueous outflow of the human TM remains to be elucidated. 2nd, we observed a consistent TGFb2-induced increase in the TEER values of the 2D HTM monolayer.17, 28 In contrast, however, the reverse TGF-b2-induced effects in the TEER, i.e., a decrease, has been reported in several studies using other sources of epithelial tissue 51-53. We currently have no explanation for why such TGF-b2-induced effects were different between HTM cells and other epithelial tissue. Nakamura et al. also studied effect of TGF and other factors on 2D HTM monolayers using TEER measurement. The conclusions reached in that study indicated that the TEER values were not statistically changed by a 24-hour exposure to TGFb1 or 254. In contrast to this, our observations consistently showed a TGFb2 induced increase in TEER values, in which the models were exposed to TGFb2 during Day1 through Day617, 28. These collective results indicate that, at a mini-mum, TGFb2 caused no change or increase in the TEER values, but not a decrease, and the longer exposure of TGFb2 may be related to differences between the study by Nakamura et al. 54 and ours. Since it is also well known that elevated levels of TGF-b2 were detected within AH obtained from patients with glaucoma. 55 It is possible that such continuous effects caused by TGF-b2 exposure may likely cause similar effects to those observed in the current study even though TGF-b2 is metabolized faster. In addition, the action of TGF-b2 on HTM cell function is currently unknown. However, since it appears that if PGF2α or OMD can actually alleviate a TGF-b2 induced increase in TEER, this rationally suggests possibility that PGF2α or OMD has some effects on TGF-b2 receptors. Thus, for a better understanding of the roles of FP or EP2 agonists within the AH dynamics of HTM, further studies to clarify above issues will be required to obtain additional insights into future therapeutic strategies for the treatment of glaucoma using PGF2α-ags and EP2 agonist.”.

Reviewer 3 Report
I have several questions regarding this manuscript.
- My first question is regarding TGF beta and TEER. Generally, TGF beta can reduce TEER, I attached several following references. However, the current paper shows TGF beta 2 can increase TEER, please give a deep discussion regarding the discrepancy between this manuscript and other literatures. In addition, do the authors add TGF beta2 daily for their studies, because TGF beta 2 has a very short turnover rate that means it can be degraded rapidly.
Schilpp C, Lochbaum R, Braubach P, Jonigk D, Frick M, Dietl P, Wittekindt OH. TGF-β1 increases permeability of ciliated airway epithelia via redistribution of claudin 3 from tight junction into cell nuclei. Pflugers Arch. 2021 Feb;473(2):287-311. doi: 10.1007/s00424-020-02501-2. Epub 2021 Jan 2. PMID: 33386991; PMCID: PMC7835204.
Nguyen N, Fernando SD, Biette KA, Hammer JA, Capocelli KE, Kitzenberg DA, Glover LE, Colgan SP, Furuta GT, Masterson JC. TGF-β1 alters esophageal epithelial barrier function by attenuation of claudin-7 in eosinophilic esophagitis. Mucosal Immunol. 2018 Mar;11(2):415-426. doi: 10.1038/mi.2017.72. Epub 2017 Aug 23. PMID: 28832026; PMCID: PMC5825237.
Li F, Pascal LE, Wang K, Zhou Y, Balasubramani GK, O'Malley KJ, Dhir R, He K, Stolz D, DeFranco DB, Yoshimura N, Nelson JB, Chong T, Guo P, He D, Wang Z. Transforming growth factor beta 1 impairs benign prostatic luminal epithelial cell monolayer barrier function. Am J Clin Exp Urol. 2020 Feb 25;8(1):9-17. PMID: 32211449; PMCID: PMC7076294.
- I would also like to ask the authors to check whether there is any tight junction proteins, such as claudins in the immortalized HTM cells, because claudins are the major components of tight junctions, and they are key elements of TEER.
- The action of TGF beta2 on HTM cell function is dependent on its receptors, and it appears that PGF2α or OMD can alleviate beta-2 induced TEER increase, does PGF2α or OMD have any effect on TGF beta2 receptors?
- Please change” Biosystems/Thermo Fisher Scientific) were performed as describe previously 17 to “Biosystems/Thermo Fisher Scientific) were performed as described previously 17”.
- My last question is regarding the TEER measurements using HTM cells 2D culture. From the following reference of Drs. Epstein and Rao’s work, the TEER value of HTM cell monolayers was much lower than the TEER value of current manuscript, I also copied their paper’s TEER value, please explain why there was so huge difference in the TEER values.
Pattabiraman, P. P., Epstein, D. L., & Rao, P. V. (2013). Regulation of Adherens Junctions in Trabecular Meshwork Cells by Rac GTPase and their influence on Intraocular Pressure. Journal of ocular biology, 1(1), 0002. https://doi.org/10.13188/2334-2838.1000002
Table 1
Effects of PDGF, H2O2 and Rac inhibitor on Transendothelial Electrical Resistance (TEER) of HTM cell monolayers.
|
Net resistance (in ohm* cm2) at various time points |
|||||||
|
Baseline |
4 hr |
6 hr |
8 hr |
12 hr |
24 hr |
48 hr |
|
|
Control |
7.7±0.8 |
7.6±0.9 |
6.9±1.1 |
6.9±0.9 |
6.9±1.2 |
6.9±0.9 |
7.0±1.1 |
|
NSC23766 |
6.0±1.3 |
5.7±1.1 |
5.8±1.1 |
3.9±0.8 * |
5.2±0.8 |
5.8±0.9 |
4.8±1.0 |
|
NAC |
8.0±1.6 |
8.0±1.2 |
5.7±1.3* |
5.1±1.0* |
5.7±1.0* |
5.7±1.0* |
5.4±1.0* |
|
H2O2 |
8.0±1.5 |
8.8±1.3 |
9.9±1.5* |
11.0±1.4* |
10.5±1.4* |
9.8±1.6* |
9.9±1.1 * |
|
PDGF |
8.3±0.9 |
9.2±0.9 |
11.1±1.3* |
10.7±1.2 * |
11.3±1.4 * |
11.7±1.4* |
10.0±1.2* |
|
NAC+PDGF |
8.3±1.0 |
8.3±1.3 |
5.3±1.2* |
5.7±0.8 * |
5.3±1.0* |
5.3±1.4* |
5.3±1.2* |
|
NAC+ H2O2 |
8.7±2.0 |
8.5±1.8 |
5.3±1.5* |
5.3±1.2* |
5.7±1.4* |
5.3±1.0* |
5.7±1.4* |
(Values are Mean±SD of three independent analyses)
*p<0.05
Thanks for the invitation.
Author Response
I have several questions regarding this manuscript.
- My first question is regarding TGF beta and TEER. Generally, TGF beta can reduce TEER, I attached several following references. However, the current paper shows TGF beta 2 can increase TEER, please give a deep discussion regarding the discrepancy between this manuscript and other literatures. In addition, do the authors add TGF beta2 daily for their studies, because TGF beta 2 has a very short turnover rate that means it can be degraded rapidly.
- Schilpp C, Lochbaum R, Braubach P, Jonigk D, Frick M, Dietl P, Wittekindt OH. TGF-β1 increases permeability of ciliated airway epithelia via redistribution of claudin 3 from tight junction into cell nuclei. Pflugers Arch. 2021 Feb;473(2):287-311. doi: 10.1007/s00424-020-02501-2. Epub 2021 Jan 2. PMID: 33386991; PMCID: PMC7835204.
- Nguyen N, Fernando SD, Biette KA, Hammer JA, Capocelli KE, Kitzenberg DA, Glover LE, Colgan SP, Furuta GT, Masterson JC. TGF-β1 alters esophageal epithelial barrier function by attenuation of claudin-7 in eosinophilic esophagitis. Mucosal Immunol. 2018 Mar;11(2):415-426. doi: 10.1038/mi.2017.72. Epub 2017 Aug 23. PMID: 28832026; PMCID: PMC5825237.
- Li F, Pascal LE, Wang K, Zhou Y, Balasubramani GK, O'Malley KJ, Dhir R, He K, Stolz D, DeFranco DB, Yoshimura N, Nelson JB, Chong T, Guo P, He D, Wang Z. Transforming growth factor beta 1 impairs benign prostatic luminal epithelial cell monolayer barrier function. Am J Clin Exp Urol. 2020 Feb 25;8(1):9-17. PMID: 32211449; PMCID: PMC7076294.
Answer; Thank you for these interesting comments. We were not aware of these discrepancies between our results of the TGFb2-induced increase of the TEER values as compared to other studies in which the TEER values are decreased by TGF. Nakamura et al. (Molecular Vision 2021;27:61-77) also studied the effect of TGF and other factors toward 2D HTM monolayer using TEER measurement, and reported that the TEER values were not statistically changed by a 24 hour exposure of TGFb1 or 2. In contrast to this, our observations of the TGFb2 induced increase in the TEER values, in which TGFb2 was exposed during Day1 through Day6, were consistently observed (Establishment of appropriate glaucoma models using dexamethasone or TGFβ2 treated three-dimension (3D) cultured human trabecular meshwork (HTM) cells. Sci Rep. 2021 Sep 29;11(1):19369. doi: 10.1038/s41598-021-98766-3; Diverse effects of pan-ROCK and ROCK2 inhibitors on 2 D and 3D cultured human trabecular meshwork (HTM) cells treated with TGFβ2. Sci Rep. 2021 Jul 27;11(1):15286. doi: 10.1038/s41598-021-94791-4; ROCK inhibitors beneficially alter the spatial configuration of TGFβ2-treated 3D organoids from a human trabecular meshwork (HTM). Sci Rep. 2020 Nov 20;10(1):20292. doi: 10.1038/s41598-020-77302-9). Taken together, at least, TGFb2 caused no change or increase in the TEER values, but no decrease, and the longer exposure of TGFb2 may be related to the difference between the study by Nakamura et al. and ours. Since it is well known that the levels of TGFb2 are elevated within aqueous humors obtained from patients with glaucoma (Targeting Transforming Growth Factor-beta Signaling in Primary Open-Angle Glaucoma.J Glaucoma. 2017 Apr;26(4):390-395. doi: 10.1097/IJG.0000000000000627), such continuous effects by TGF-2 exposure may likely cause similar effects, as was observed in the current study, even though TGF-b2 is metabolized faster. Therefore, we believe that our longer exposure model is a more representative model. Therefore, this information is included in the discussion.
- I would also like to ask the authors to check whether there is any tight junction proteins, such as claudins in the immortalized HTM cells, because claudins are the major components of tight junctions, and they are key elements of TEER.
Answer; Thank you for this constructive comment. We are in compete agreement with you that cell adhesion factors including claudins and others should be important in understanding the current TEER results. However, since human TM is composed of multiple sheet layers rather than a monolayer, we therefore assume that the 3D spheroid model is also important for understanding the drug induced effects toward our 2D and 3D HTM cultured models, in addition to the analysis of TEER of the HTM monolayer. Therefore, this information is now included in the study limitation in the Discussion.
- The action of TGF beta2 on HTM cell function is dependent on its receptors, and it appears that PGF2α or OMD can alleviate beta-2 induced TEER increase, does PGF2α or OMD have any effect on TGF beta2 receptors?
Answer; Thank you for this excellent idea for understanding drug induced effects toward TGF-b2 treated our HTM models. Therefore this was also included in the study limitations in the Discussion section; “To elucidate the underlying mechanisms responsible for causing such diverse effects between PGF2α and OMD as above, we evaluated the gene expressions of several molecules that might possibly be determinants of the structures of the 2D and 3D cultured HTM cells. These studies included ECMs, ECM regulatory factors, TIMPs and MMPs, and ER-stress related factors (Figs. 4-7). However, as summarized in Table 1, the mRNA ex-pressions of these factors within the 2D and 3D cultured HTM cells were not significantly different between PGF2α and OMD. However, the current study has the following limitation that need to be taken into consideration; 1st, the relationship between such collective observations and the ease of aqueous outflow of the human TM remains to be elucidated. 2nd, we observed a consistent TGFb2-induced increase in the TEER values of the 2D HTM monolayer.17, 28 In contrast, however, the reverse TGF-b2-induced effects in the TEER, i.e., a decrease, has been reported in several studies using other sources of epithelial tissue 51-53. We currently have no explanation for why such TGF-b2-induced effects were different between HTM cells and other epithelial tissue. Nakamura et al. also studied effect of TGF and other factors on 2D HTM monolayers using TEER measurement. The conclusions reached in that study indicated that the TEER values were not statistically changed by a 24-hour exposure to TGFb1 or 254. In contrast to this, our observations consistently showed a TGFb2 induced increase in TEER values, in which the models were exposed to TGFb2 during Day1 through Day617, 28. These collective results indicate that, at a mini-mum, TGFb2 caused no change or increase in the TEER values, but not a decrease, and the longer exposure of TGFb2 may be related to differences between the study by Nakamura et al. 54 and ours. Since it is also well known that elevated levels of TGF-b2 were detected within AH obtained from patients with glaucoma. 55 It is possible that such continuous effects caused by TGF-b2 exposure may likely cause similar effects to those observed in the current study even though TGF-b2 is metabolized faster. In addition, the action of TGF-b2 on HTM cell function is currently unknown. However, since it appears that if PGF2α or OMD can actually alleviate a TGF-b2 induced increase in TEER, this rationally suggests possibility that PGF2α or OMD has some effects on TGF-b2 receptors. Thus, for a better understanding of the roles of FP or EP2 agonists within the AH dynamics of HTM, further studies to clarify above issues will be required to obtain additional insights into future therapeutic strategies for the treatment of glaucoma using PGF2α-ags and EP2 agonist.”.
- Please change” Biosystems/Thermo Fisher Scientific) were performed as describe previously 17 to “Biosystems/Thermo Fisher Scientific) were performed as described previously 17”.
Answer; As pointed out, “describe” was changed to “described”.
- My last question is regarding the TEER measurements using HTM cells 2D culture. From the following reference of Drs. Epstein and Rao’s work, the TEER value of HTM cell monolayers was much lower than the TEER value of current manuscript, I also copied their paper’s TEER value, please explain why there was so huge difference in the TEER values.
Answer; Thank you for this comment. Regarding the difference in the TEER values, we have no explanation for this. However, as a possibility, we assumed that 1) the instrument used for the TEER measurements including a membrane with permeable pores (ours; 0.4 mm pore, Epstein and Rao; 0.4 m pore polycarbonate membrane #3407, Costar), and other devices, 2) HTM cells may be different (ours were commercially available immortalized HTM, Epstein and Rao; HTM cells obtained from donors, age 77, 64 and 55 years). However, despite the basal TEER values, it is possible that the difference in the TEER values can be attributed to the experimental conditions.

Round 2
Reviewer 1 Report
I am afraid, most of the answers and added test did not properly addressed my concerns.
- Language is definitely improved since last time; thanks to dr. Dr. Feather. However, i am not sure the new part highlighted in yellow received similar attention as the rest of the manuscript. This part still present important spelling and grammar mistakes and it is sometime difficult to read.
- The model used is still referred to as an "in vivo" model. Authors attempted to explain the reason of this choice yet consensus is that such a model employing isolated cells should be considered an "in vitro" model. This is not a minor concern; authors cannot sell something that they do not have as it will result misleading and confusing to the audience - to be honest, this attitde is really disappointing.
- We mentioned that statistics needed some revision. This is another very important aspect still not addressed in the new version. Authors refer to t-test within their explanation while all figures state that ANOVA followed by a Tukey’s multiple comparison was used; there is no doubt the statistics applied has important flaws raising additional doubts and concerns on the overall accuracy of the results presented.
- Authors seems to be aware of the main MoA of PGF2alpha and EP2 agonists being it mostly on uveoscleral outflow yet they use these drugs in a model of conventional outflow; ok, authors claim that their intention was to address potential activity of these compounds on TM system which is reasonable but than a positive control should be included and still the discussion of the data requires attention towards the main MoA of these drugs. Again, not clearly stating that within introduction, results and discussion would result mis-leading and confusing.
Author Response
Dear Editor,
Thank you very much again for the constructive comments concerning our manuscript; " Comparison of the drug induced effects of a selective EP2 agonist, Omidenepag isopropyl, with PGF2α, on TGF-β2-treated human trabecular meshwork (HTM) cells “. We carefully examined all of the comments made by the Reviewer and made a series of specific changes to our manuscript as follows;
Reviewer1
- I am afraid, most of the answers and added test did not properly addressed my concerns.
Answer; I deeply apologize because I did not fully understand your raised questions and therefore there were some my mis-understanding. In this time, I do my best to answer you raised questions below.
- Language is definitely improved since last time; thanks to dr. Dr. Feather. However, i am not sure the new part highlighted in yellow received similar attention as the rest of the manuscript. This part still present important spelling and grammar mistakes and it is sometime difficult to read.
Answer; Thank you for this comment. As suggested, I asked Dr. Feather to check the new part highlighted in yellow as well as several other new sentences modified according following questions.
- The model used is still referred to as an "in vivo" model. Authors attempted to explain the reason of this choice yet consensus is that such a model employing isolated cells should be considered an "in vitro" model. This is not a minor concern; authors cannot sell something that they do not have as it will result misleading and confusing to the audience - to be honest, this attitde is really disappointing.
Answer; Again, I am so sorry for my answer to this question. I was totally mis-understood and overstated. I agree with your suggestion that even if our 3D cell culture has some advantages in terms of estimating in vivo functions compared to the conventional 2D cell culture, our 3D culture should still be in vitro system, but not in vivo system. Therefore, I changed “in vivo” to “in vitro”.
- We mentioned that statistics needed some revision. This is another very important aspect still not addressed in the new version. Authors refer to t-test within their explanation while all figures state that ANOVA followed by a Tukey’s multiple comparison was used; there is no doubt the statistics applied has important flaws raising additional doubts and concerns on the overall accuracy of the results presented.
Answer; Thank you for this question. Again, I misunderstood your question last time, and performed wrong additional evaluation by t-test. Therefore, as pointed out, that led to further doubts as suggested. Therefore, I would like to answer again to the initial your question; “In many occasions, statistics seem inappropriate as it revels highly significant values even if changes are rather modest and errors overlap i.e. fig 2 and fig.3.”, as follow. To show that the statistics are, in fact, appropriate, especially in Figs, 2 and 3, I attached the corresponding PDFs of the excel data showing the experimentally observed values, and a statistical analysis of the data. In addition, other all data were again checked very carefully and have now been corrected.
- Authors seems to be aware of the main MoA of PGF2alpha and EP2 agonists being it mostly on uveoscleral outflow yet they use these drugs in a model of conventional outflow; ok, authors claim that their intention was to address potential activity of these compounds on TM system which is reasonable but than a positive control should be included and still the discussion of the data requires attention towards the main MoA of these drugs. Again, not clearly stating that within introduction, results and discussion would result mis-leading and confusing.
Answer; Thank you for this question. I agree with this pointed-out issue that the study rationale for evaluating these drug effects toward the conventional AH out flow rather than uveoscleral outflow. To clarify this ambiguity, we emphasized the need to study the inconsistency between the ratios of the responsible conventional (80-90%) and uveoscleral routes (10-20%) of AH drainage, and the powerful hypotensive efficacy of both drugs, as our current study rationale to avoid mis-leading. In addition, as suggested, our recent study using the ROCK inhibitors on the same 2D and 3D HTM system (Sci Rep. 2020 Nov 20;10(1):20292. Sci Rep. 2021 Jul 27;11(1):15286.) was included as a positive control, because it is well known that ROCK inhibitors are established effecters toward the conventional AH outflow route (Jpn J Ophthalmol 2018;62, 109-126, Expert Opin Ther Pat 2019; 29, 817-827). Therefore, this information is now included in the rewritten 2nd paragraph of the introduction; “Among the current clinically available hypotensive anti-glaucoma medications, prostaglandin (PG) F2α agonists (PGF2α-ags) are first-line drugs for the treatment of GON [10-12]. In addition to PGF2α-ags, a non-prostaglandin, a prostanoid EP2 agonist Omidenepag isopropyl (OMDI), which is converted into the active form (Omidenepag, OMD) by spontaneous hydrolysis after its administration has recently been made available for the treatment of patients with OH and POAG [13,14]. It is generally thought that the hypotensive mechanisms for both PGF2α-ags and the EP2 agonist, OMD, involve increasing the ease of uveoscleral outflow, but not the ease of conventional outflow [15]. It is well recognized that approximately 80-90 % of AH drains through the conventional outflow route, while the remaining 10-20 % is excreted via the uveoscleral outflow route [16]. Despite this, no rationale reasons have been proposed to explain precisely why IOPs are decreased by both PGF2α-ags and the EP2 agonist despite the fact that AH drainage through the uveoscleral outflow route is only 10 – 20%. Therefore, these collective observations strongly suggest that both PGF2α-ags and the EP2 agonist may also affect the conventional route in addition to the uveoscleral route. However, the effects of these drugs on the ease of conventional outflow have not been extensively characterized at this time, since an appropriate experimental model for evaluating the drug efficacies on conventional versus uveoscleral roots has not been available. More interestingly, an in vivo study using OH monkeys reported that the dynamics of the pharmacokinetics of OMD in the AH are quite different from those of PGF2α-ags [17]. Therefore, although both PGF2α-ags and OMD affect AH outflow by the uveoscleral root in similar manners and are categorized as the same members of PG-related compounds, the effect of OMD may be different from that for PGF2α-ags toward human TM, especially glaucomatous human TM. To address this issue, we recently developed an in vitro model that replicates the human TM (HTM) using a unique 3D drop culture method [18], in which no assistance such as a collagen-gel or a scaffold that are usually used in the 3D culture methods [19] were required. In fact, using this in vitro 3D HTM spheroid model, we recently confirmed the effectiveness the non-selective pan-Rho-associated coiled-coil containing protein kinases (ROCKs) inhibitors, ripasudil hydrochloride hydrate (Rip) and Y27632 [18,20] on TGF-β2 treated HTM cells. Both of the above have been established effectors toward the conventional route [21][22]. In addition, when using this model for steroid-induced glaucoma in an evaluation of PGF2α-ags and OMD, significant and diverse effects were detected in dexamethasone treated HTM cells [23], in addition to adipogenesis of the 3T3-L1 mouse preadipocytes [24,25] as well as human orbital fibroblasts (HOF) [26] and Graves’ orbitopathy related HOF[27].”, 1st sentence in the 2nd paragraph of result; “Using this strategy, we investigated the unidentified effects of PGF2α and the selective EP2 agonist, omidenepag (OMD) on the conventional AH drainage route, especially human TM, we initially compared the effect of these drugs on TGF-β2 treated HTM cells. In this study, the barrier function of 2D cultured HTM cell monolayers was measured by transendothelial electron resistance (TEER).”, and last paragraph of discussion; “In conclusion, the findings presented in this study indicate that PGF2α and OMD significantly and differently modulated the TEER values of TGF-β2 treated 2D HTM monolayers as well as the physical properties of the TGF-β2 treated 3D HTM spheroids, suggesting that both may also affect, not only the uveoscleral AH outflow route, but conventional outflow as well.”.

Reviewer 3 Report
The authors revised the manuscript and addressed most of the points I have, however, there is one more thing the authors needs to pay special attention and address it with caution, that is why the authors have much higher TEER reading in 2D HTM culture than others . Please explain why there was so huge difference in the TEER values.
I listed 3 examples to explain it.
- It is well established that the major flow resistance should be generated at the inner wall of the Schlemm’s canal (SC) region, but not in the TM region, because SC cells generally contain some tight junctions, however, according to Dr. Stamer’s 2012 paper, the SC cells only have about 16 ohm*cm2 TEER value, and the present manuscript have at least 5 times higher TEER reading in their HTM monolayer cells.
Perkumas KM, Stamer WD. Protein markers and differentiation in culture for Schlemm's canal endothelial cells. Exp Eye Res. 2012 Mar; 96(1):82-7. doi: 10.1016/j.exer.2011.12.017. Epub 2011 Dec 22. PMID: 22210126; PMCID: PMC3296889.
Fig. 4. Effect of growth factor withdrawal on transendothelial electrical resistance (TEER) and vascular endothelial (VE) cadherin expression by Schlemm’s canal endothelia over time. SC cell monolayers on filters were switched from fetal bovine serum (FBS) to adult bovine serum (ABS) after one week at confluence. Mature monolayers of SC cells were tested for TEER (panel A) or expression of the adherens protein, VE-cadherin (panels B) following change to media containing ABS every two days. Panel C shows data examining change in VE-cadherin protein in 4 different SC cell strains 6 days after switch to ABS, the time point showing first significant difference in TEER. Significant differences (p < 0.05) in both VE-cadherin expression and TEER were detected following treatment with ABS (n = 3–5 experiments using 4 different cell strains).
- From Drs. Epstein and Rao’s work, the TEER value of HTM cell monolayers was much lower than the TEER value of current manuscript, I also copied their paper’s TEER value.
Pattabiraman, P. P., Epstein, D. L., & Rao, P. V. (2013). Regulation of Adherens Junctions in Trabecular Meshwork Cells by Rac GTPase and their influence on Intraocular Pressure. Journal of ocular biology, 1(1), 0002. https://doi.org/10.13188/2334-2838.1000002
Table 1
Effects of PDGF, H2O2 and Rac inhibitor on Transendothelial Electrical Resistance (TEER) of HTM cell monolayers.
|
Net resistance (in ohm* cm2) at various time points |
|||||||
|
Baseline |
4 hr |
6 hr |
8 hr |
12 hr |
24 hr |
48 hr |
|
|
Control |
7.7±0.8 |
7.6±0.9 |
6.9±1.1 |
6.9±0.9 |
6.9±1.2 |
6.9±0.9 |
7.0±1.1 |
|
NSC23766 |
6.0±1.3 |
5.7±1.1 |
5.8±1.1 |
3.9±0.8 * |
5.2±0.8 |
5.8±0.9 |
4.8±1.0 |
|
NAC |
8.0±1.6 |
8.0±1.2 |
5.7±1.3* |
5.1±1.0* |
5.7±1.0* |
5.7±1.0* |
5.4±1.0* |
|
H2O2 |
8.0±1.5 |
8.8±1.3 |
9.9±1.5* |
11.0±1.4* |
10.5±1.4* |
9.8±1.6* |
9.9±1.1 * |
|
PDGF |
8.3±0.9 |
9.2±0.9 |
11.1±1.3* |
10.7±1.2 * |
11.3±1.4 * |
11.7±1.4* |
10.0±1.2* |
|
NAC+PDGF |
8.3±1.0 |
8.3±1.3 |
5.3±1.2* |
5.7±0.8 * |
5.3±1.0* |
5.3±1.4* |
5.3±1.2* |
|
NAC+ H2O2 |
8.7±2.0 |
8.5±1.8 |
5.3±1.5* |
5.3±1.2* |
5.7±1.4* |
5.3±1.0* |
5.7±1.4* |
(Values are Mean±SD of three independent analyses)
*p<0.05
- We have also tried to measure TEER in our HTM monolayer cells, however, we can’t get a positive reading number after subtraction of the medium only TEER control reading. We used HUVECs and MDCK cells as positive control, we can get positive reading in these cells with TEERs in HUVECs around 100 ohm* cm2, and TEER in MDCKs much higher than in HUVECs.
Thanks for the invitation.
Author Response
Dear Editor,
Thank you very much again for the constructive comments concerning our manuscript; " Comparison of the drug induced effects of a selective EP2 agonist, Omidenepag isopropyl, with PGF2α, on TGF-β2-treated human trabecular meshwork (HTM) cells “. We carefully examined all of the comments made by the Reviewer and made a series of specific changes to our manuscript as follows;
Reviewer 3
The authors revised the manuscript and addressed most of the points I have, however, there is one more thing the authors needs to pay special attention and address it with caution, that is why the authors have much higher TEER reading in 2D HTM culture than others. Please explain why there was so huge difference in the TEER values.
I list 3 examples to explain this.
- It is well established that the major flow resistance should be generated at the inner wall of the Schlemm’s canal (SC) region, but not in the TM region, because SC cells generally contain some tight junctions, however, according to Dr. Stamer’s 2012 paper, the SC cells only have about 16 ohm*cm2 TEER value, and the present manuscript have at least 5 times higher TEER reading in their HTM monolayer cells.
Perkumas KM, Stamer WD. Protein markers and differentiation in culture for Schlemm's canal endothelial cells. Exp Eye Res. 2012 Mar; 96(1):82-7. doi: 10.1016/j.exer.2011.12.017. Epub 2011 Dec 22. PMID: 22210126; PMCID: PMC3296889. Fig. 4. Effect of growth factor withdrawal on transendothelial electrical resistance (TEER) and vascular endothelial (VE) cadherin expression by Schlemm’s canal endothelia over time. SC cell monolayers on filters were switched from fetal bovine serum (FBS) to adult bovine serum (ABS) after one week at confluence. Mature monolayers of SC cells were tested for TEER (panel A) or expression of the adherens protein, VE-cadherin (panels B) following change to media containing ABS every two days. Panel C shows data examining change in VE-cadherin protein in 4 different SC cell strains 6 days after switch to ABS, the time point showing first significant difference in TEER. Significant differences (p < 0.05) in both VE-cadherin expression and TEER were detected following treatment with ABS (n = 3–5 experiments using 4 different cell strains).
- From Drs. Epstein and Rao’s work, the TEER value of HTM cell monolayers was much lower than the TEER value of current manuscript, I also copied their paper’s TEER value.
Pattabiraman, P. P., Epstein, D. L., & Rao, P. V. (2013). Regulation of Adherens Junctions in Trabecular Meshwork Cells by Rac GTPase and their influence on Intraocular Pressure. Journal of ocular biology, 1(1), 0002. https://doi.org/10.13188/2334-2838.1000002
Table 1
Effects of PDGF, H2O2 and Rac inhibitor on Transendothelial Electrical Resistance (TEER) of HTM cell monolayers.
|
Net resistance (in ohm* cm2) at various time points |
|||||||
|
|
Baseline |
4 hr |
6 hr |
8 hr |
12 hr |
24 hr |
48 hr |
|
Control |
7.7±0.8 |
7.6±0.9 |
6.9±1.1 |
6.9±0.9 |
6.9±1.2 |
6.9±0.9 |
7.0±1.1 |
|
NSC23766 |
6.0±1.3 |
5.7±1.1 |
5.8±1.1 |
3.9±0.8 * |
5.2±0.8 |
5.8±0.9 |
4.8±1.0 |
|
NAC |
8.0±1.6 |
8.0±1.2 |
5.7±1.3* |
5.1±1.0* |
5.7±1.0* |
5.7±1.0* |
5.4±1.0* |
|
H2O2 |
8.0±1.5 |
8.8±1.3 |
9.9±1.5* |
11.0±1.4* |
10.5±1.4* |
9.8±1.6* |
9.9±1.1 * |
|
PDGF |
8.3±0.9 |
9.2±0.9 |
11.1±1.3* |
10.7±1.2 * |
11.3±1.4 * |
11.7±1.4* |
10.0±1.2* |
|
NAC+PDGF |
8.3±1.0 |
8.3±1.3 |
5.3±1.2* |
5.7±0.8 * |
5.3±1.0* |
5.3±1.4* |
5.3±1.2* |
|
NAC+ H2O2 |
8.7±2.0 |
8.5±1.8 |
5.3±1.5* |
5.3±1.2* |
5.7±1.4* |
5.3±1.0* |
5.7±1.4* |
(Values are the Mean±SD of three independent analyses)
*p<0.05
- We have also tried to measure TEER in our HTM monolayer cells, however, we can’t get a positive reading number after subtraction of the medium only TEER control reading. We used HUVECs and MDCK cells as positive control, we can get positive reading in these cells with TEERs in HUVECs around 100 ohm* cm2, and TEER in MDCKs much higher than in HUVECs.
Answer; Thank you so much for pointing out this very important issue regarding the TEER values. I check the experimental procedures described in our Lab notebook very carefully and we discovered that we used the wrong control. TEER values were measured with the well just filled with PBS, instead of being initially incubated with the culture medium and then washed with PBS twice, the same as the sample measurements. Therefore, TEER results were corrected by taking into account the use of the proper control (approximately 85 Ωcm2). We replaced with new corrected figure, and recalculated the statistical analysis. However, the obtained results were essentially unchanged. We are very sorry for this, and we wish to thank you again for your constructive suggestions.
